# Specific Radar Recognition Based on Characteristics of Emitted Radio Waveforms Using Convolutional Neural Networks

**DOI:** 10.3390/s21248237

**Published:** 2021-12-09

**Authors:** Jan Matuszewski, Dymitr Pietrow

**Affiliations:** Institute of Radioelectronics, Faculty of Electronics, Military University of Technology, 00-908 Warsaw, Poland; dymitr.pietrow@wat.edu.pl

**Keywords:** convolutional neural networks, radar recognition, deep learning, signal simulation, electronic warfare

## Abstract

With the increasing complexity of the electromagnetic environment and continuous development of radar technology we can expect a large number of modern radars using agile waveforms to appear on the battlefield in the near future. Effectively identifying these radar signals in electronic warfare systems only by relying on traditional recognition models poses a serious challenge. In response to the above problem, this paper proposes a recognition method of emitted radar signals with agile waveforms based on the convolutional neural network (CNN). These signals are measured in the electronic recognition receivers and processed into digital data, after which they undergo recognition. The implementation of this system is presented in a simulation environment with the help of a signal generator that has the ability to make changes in signal signatures earlier recognized and written in the emitter database. This article contains a description of the software’s components, learning subsystem and signal generator. The problem of teaching neural networks with the use of the graphics processing units and the way of choosing the learning coefficients are also outlined. The correctness of the CNN operation was tested using a simulation environment that verified the operation’s effectiveness in a noisy environment and in conditions where many radar signals that interfere with each other are present. The effectiveness results of the applied solutions and the possibilities of developing the method of learning and processing algorithms are presented by means of tables and appropriate figures. The experimental results demonstrate that the proposed method can effectively solve the problem of recognizing raw radar signals with agile time waveforms, and achieve correct probability of recognition at the level of 92–99%.

## 1. Introduction

In radio-electronic reconnaissance systems we receive and then measure the basic time, frequency and spatial parameters (related to the scanning of the antenna) in order to recognize their emission sources (in our case, radars), and we do not visualize the spatial situation using signals. The radio-electronic reconnaissance systems extract the basic characteristic parameters from measured radar signals. Based on these parameters, we can obtain information such as the system, application, type and platform of the radar, and further deduce the battlefield situation, threat level, activity rule, tactical intention, etc., and provide important intelligence support for our own decision-making system. The modern electromagnetic environment is considered complex due to a multitude of signals originating from a number of different radars (emitters), and in the case of signals coming from the same radar their parameters (features) are measured with low accuracy. In many cases the radar may change one or several signal parameters in order to perform its task more efficiently [1,2]. Since each radar has limited parameter ranges (e.g., transmits within a limited frequency band) and often identifiable characteristics, it is assumed that radar signals with similar characteristics originate from the same device [3,4].

Artificial neural networks (ANNs) currently achieve high efficiency in the field of signal and image processing, especially in the field of different objects recognition [5,6,7,8,9,10] and extending situational awareness supporting, for example, SAR imaging [11,12,13]. Interest in the implementation of artificial intelligence methods in the field of broadly understood classification is constantly growing [14,15,16,17,18,19,20,21,22,23,24,25,26,27,28], which is caused by the continuous improvement of learning algorithms and the increase in the computing power of computers [29,30,31,32,33,34]. The conventional approach to digital signals processing with the use of convolutional artificial neural networks (CNNs) is the signal transition from the time domain to the time-frequency domain, i.e., to the signal image [35,36,37,38], which takes into account a typical structure of the convolutional neural network (CNN) [25,26,39], which is much easier to process than the raw signal.

Opponents of artificial intelligence methods, especially deep learning methods, will point out here that the learning process may take too long in relation to the use of manually programmed filters/processing algorithms. However, in the case of deep neural networks, it is possible to use already learned filters from convolutional layers [40,41,42,43,44]. The only problem that remains then is teaching a three-layer neural network, which is usually learned quite quickly in relation to convolutional layers [35,42,45,46,47]. Another cause for optimism is that modern graphics cards make it possible to reduce the training time of ANN classifiers from several weeks to hours or even minutes (here the leaders in this field are NVIDIA and AMD) [30,31,48]. The ANN processing methods discussed in this article will operate on digital signals, and will show the possibility of optimizing the methods of classifying these radar signals depending on the receiver that produces a given signal, as well as the acceleration of the processing itself. The nature of ANN structures and their operation can be used for data processing, and for signals at higher frequencies directly or indirectly (after rescale or transforming high-frequency signals to the lower baseband frequency) in which the telecommunications or radiolocation devices operate [8,39,49].

In our work we chose deep learning (DL) with convolutional neural networks for signal processing because we wanted to create an agile and adjustable (to radio background) radar signal recognition system [50,51,52]. DL for signal processing can be superior in comparison to the standard models of neural networks with full connections (i.e., as used in [53]) or processing with hand-crafted features (e.g., eigenvector analysis methods [54], random forest (RF), support vector machines (SVM), k-nearest neighbors [55]), because they do not require neither the use of preliminary signal processing, nor additional feature extraction, nor the standardization of measured vectors. The neural network acquires all the features of extraction during the learning process. However, as it is stated in other publications [55,56], to achieve better results in comparison to algorithms with hand-crafted extraction features we need enough data patterns to learn the deep model otherwise we do not achieve better performance but can lead the deep model to overfitting. A better approach to improve system performance can be achieved by supporting the DL method with hand-crafted pre-processing so we can speed up the learning process and outperformance application of each method separately [55]. In our work such preliminary processing can be called the use of the FIR filter before the entry of the classifier. Without the FIR filter, the convolutional neural network would still be able to classify the raw signals, except that we would probably have to add one layer or two to the network in order to teach it how to filter the desired signals from the frequency domain.

In paper [54] eigenvector analysis is used, which is quite efficient, however, as the number of analyzed devices increases, the eigenvector matrix grows, and this can gradually lead to deteriorating system performance. In addition, the methods based on the analysis of eigenvectors degrade some information when converting the classifying vectors to the matrix of eigenvectors. The degradation of some information may lead to deterioration of the classification under disturbance conditions or when geometrical changes of the input samples are inserted.

When using artificial neural networks (especially convolutional networks), adding further radar classifying vectors, it is not required to enlarge the structure of the neural network, but we can consider use of learning transfer, which means disconnecting the last layer of the neural network and connecting another one with the number of required classes and training this layer with the use of the pre-trained first layers [56]. Another argument regarding the use of convolutional neural networks is that there is no need to lose information in the pre-processing signals in the form of reducing the dimensions or averaging the available processing channels. The network, during the learning process through the use of layers such as convolutional and max pooling, decides on the basis of the training set which signal features are unimportant and what will be relevant in the process of proper classification [56].

It also should be noted that in the methods shown in other works [53,54] we process only post-detection data without introducing any significant disturbances. In the work presented here we process only the raw signals without complicated pre-processing, which is mostly carried out by layers of the neural network. Processing signals of the size 355 × $10^{3}$ samples with the use of the method of eigenvector analysis would force us to create huge matrices, the calculation of which in real time may sometimes prove impossible.

In the following parts of this article, Section 2 contains a description of an electromagnetic environment, the measurement method and the processing of radar signal parameters, as well as the method of constructing the training dataset applied in the proposed model for recognizing emitter sources. Section 3 presents a detailed structure of CNN designed for radar signals recognition based on the measured parameters, i.e., pulse repetition interval, pulse duration, radio frequency and antenna scan period. The learning method designed for a CNN structure, the software used and the mathematical formulas are described in Section 4. The simulation environment used for testing the efficiency of radar signal recognition is described in Section 5. Analysis of experiment results, shown in the form of tables and figures, is presented in Section 6. Section 7 contains a summary of the obtained simulation results and the conclusion.

## 2. Description of the Database

### 2.1. Description of Radar Signal Parameters

The electronic support measures (ESM) systems measure the basic parameters of incoming radar signals (frequency, amplitude, bearing and elevation angles, pulse width, time of arrival and sometimes polarization). The data collected are sorted into groups considered to be from a single emitter and subsequently used to compute the time-dependent parameters (pulse repetition frequency, antenna rotation period, etc.) [1,2,3]. Finally, the system matches the “signal signatures”, composed from the average parameters from each group, with the characteristics of known emitters stored in the emitter database (EDB). This action enables the system to identify and classify the incoming radar signals which may have a high degree of inherent uncertainty arising from the methods of data gathering and processing [2,3]. In the electromagnetic environment a great deal of information collected by the receiver is processed in real time (Figure 1).

The basic measured radar signals in mobile systems for identification of radar signals MUR-20 or in the ELINT system of recognition of onboard RF emitters, produced by PIT-RADWAR, are the following:-Automatic detection direction that finds and monitors the emission sources with a frequency ranging from 500 MHz to 18 GHz;-Signal parameters measured: frequency, pulse width, amplitude, direction of arrival, pulse repetition frequency, antenna rotation period;-Deinterleaving;-Acousto-optical channel of spectrum analyzer 500 MHz and channel of compression spectrum analyzer 40 MHz;-Radio frequency measurement with 1 MHz accuracy;-Instantaneous time parameters measurement with 25 ns accuracy.

The signal structure generated by a single radar varies across time and depends on parameters such as pulse repetition interval (PRI), pulse duration (PD), radio frequency (RF) and antenna scan period (SP). The intervals of individual radar signal parameters may overlap, therefore the given signals may be more or less similar to each other at certain times. The recognition of measured radar signal parameters is also based on the analysis of their temporal structure, which will be reconstructed using the simulation environment, because unlike the PRI, PD, RF and SP parameters, we do not have signal temporal structures assigned to particular classes of radar signals.

The radar signals characteristics presented in this section belong to 18 different types of radars (classes). Table 1 presents the confidence intervals of radar signal parameters calculated based on their measurement data.

Figure 2 and Figure 3 show the time and frequency dependencies between individual classes of signal parameters. The intervals of individual classes have been marked with different colors. The number of classes is successively marked on the right side from 0 to 17. The interval width for each class means the time or frequency interval in which the signals will subsequently be generated for each class.

### 2.2. Constructing a Set of Data for Training a Neural Network

As noted in the previous section, based on the description and parameters intervals assigned to the individual radars, these classes in certain ranges can overlap each other, which can significantly affect the effectiveness of proper signal source recognition. To reduce the risk of a false recognition, the final decision concerning the recognition of a given signal source is announced only after receiving all the parameters described above. For this purpose, three training sets were created.

(a)The first training dataset consists of time waveforms (TW) of the signals with variable PD, RF and intra-pulse modulation. An example of the time waveforms of a signal simulated with the use of a simulation environment (which is described in more detail in Section 5), based on the parameters presented in Table 1, is depicted in Figure 4.

(b)The second training dataset consists of variable PRI waveforms which change depending on the applied inter-pulse modulation. Below, in Figure 5, these changes of PRI are shown.

(c)The third training dataset consists of variable PD waveforms changing from pulse to pulse.

### 2.3. The Similarity between the Classes of Signals

From the waveforms of changes of individual parameters presented in Section 2.1 (Figure 2 and Figure 3), the overlapping of individual classes of signals can be noted, taking into account time and frequency dependencies. Table 2 below depicts the classes with overlapping ranges of changes in time (PRI, PD, SP) and frequency (RF).

## 3. Proposed Model

The CNN model applied in this paper is a multi-layer (deep) structure containing inter-area convolutional connections in the first layers, batch normalization layers [57] applied during the learning process to accelerate and stabilize it, a sub-sampling layer (max-pooling) [58] and dropout layers reducing the probability of overfitting [59]. Figure 6 below depicts the structure of the CNN used for training the signals of changeable intra-pulse modulation TW (RF and PD), PRI and PD signals with inter-pulse modulation.

The first CNN layers in this structure, in this case the convolutional layers 1D, are designed to extract the features from the signals tested [60]. Traditional convolutional 2D layers operate on images and use the 2D filters to extract features of input signal, where convolutional 1D layers have 1D filters and operate directly on the 1D signal without transformations (such as transforming signal 1D to a spectrogram). Extraction of selected signal features is carried out by the means of a convolution operation of the input signal with feature maps (filters) obtained during the learning process. Figure 7 shows examples of signal filters (visualization of weight maps) obtained during the learning process. Due to the CNN structure used together with the one-dimensional convolution layers, the illustration of these filters was presented in the form of sampled time waveforms.

In the designed structure feature extraction has been performed three times. After each extraction, the maps of neuron responses obtained at the output of the convolution layer were down-sampled with the help of the max pooling layer [58] in order to facilitate the features’ extraction in the successive convolution layers. The last layers—the dense (full connected) layers [4,61]—are the layers deciding on the basis of the extracted features which class of signals we deal with. Table 3 presents a detailed description of the structure of each layer in the CNN structure described above.

The second CNN structure uses three structures similar to the model described above. The structures are connected by concatenate layer [63] and work together simultaneously processing PRI, PD parameters and TW to determine the class of input signals. The last layer of this structure is the dense type and makes final recognition of the signal (Figure 8 and Table 4). The model described here is depicted in more detail in Figure 8.

The third CNN model is intended to recognize raw samples of vector signals. The first processing layers are comparable to structures described above, although input of the model is quite large with 335,544 samples and the last layers are transpose convolution layers [64]. Figure 9 and Table 5 presented below describe the model in more detail.

Table 6 presents the memory requirement parameters and performance while processing of single input vector data for the described above CNN architecture.

## 4. CNN Learning

The learning of CNN structures presented in this article was carried out using the TensorFlow [4] library, which enables the acceleration of the learning process using general-purpose computing in graphics processing units (GP-GPU) processors (graphics cards), the use of modern learning algorithms and constructing structures of artificial neural networks.

The learning process was carried out using Adam Optimizer, a modification of the stochastic gradient descent (SGD) algorithm [65], which allows for the efficient solving of the optimization problems for multidimensional objective functions. Due to its efficiency, this algorithm is implemented by default in ANN training libraries such as TensorFlow or Keras [66]. The Adam Optimizer gradient descent algorithm is an adaptive learning algorithm, which is an extension and combination of two methods, i.e., the AdaGrad and RMSProp methods [65]. Basic Equations (1–6) of the Adam Optimizer algorithm are presented below, according to which the successive values of changes of weight vector in the ANN are calculated in the following way:(1)g(i)=∇E(W(i)),
(2)$$m\left(i\right)=\beta_{1}\left(i\right)m\left(i-1\right)+\left(1-\beta_{1}\left(i\right)\right)g\left(i\right),$$
(3)$$v\left(i\right)=\beta_{2}\left(i\right)v\left(i-1\right)+\left(1-\beta_{2}\left(i\right)\right)g\left(i\right),$$
(4)$$\hat{m}\left(i\right)=\frac{m\left(i\right)}{1-\beta_{1}\left(i\right)},$$
(5)$$\hat{v}\left(i\right)=\frac{m\left(i\right)}{1-\beta_{2}\left(i\right)},$$
(6)$$W\left(i\right)=W\left(i-1\right)-\frac{\alpha \hat{m}\left(i\right)}{\sqrt{\hat{v}\left(i\right)}+\epsilon }$$
where i is a number of the current epoch in the learning process, g(i) is the gradient of the objective function, m(i) is the first-order moment of the estimation of changes in the value of the weight vector, v(i) is the second-order moment of the estimate of changes in the value of the weight vector, $\hat{m}\left(i\right)$ denotes the normalization of the moment m(i), $\hat{v}\left(i\right)$ denotes the normalization of the moment v(i), $\beta_{1}\left(i\right)$ is the moment decay factor m(i), $\beta_{2}\left(i\right)$ is the moment decay factor v(i), W(i) denotes the weight vector of CNN, α is the learning coefficient or the step of changes in the weight vector updating, and ϵ is the small value ensuring stability of calculations.

In the first step the Adam Optimizer program calculates the gradient of the objective function (multivariate error function) g(i) in Equation (1). Then, respectively, the first m(i) and second-order moments v(i) are calculated. These are the values of the estimation of the weight vector changes W(i) in Equations (2) and (3). Before the actual updating of the weights vector, Equation (6), the correction (normalization) of the moments v(i) and m(i) is performed. During each update, along with the learning progress, the effects of the moments of the first and second order of the estimates are minimized on the basis of the $\beta_{1}$ and $\beta_{2}$ coefficients.

Due to the fact that the presentation of the learning patterns is carried out using an indirect method between online (updating weights after presenting one learning pattern) or total batch (updating weights after presenting the entire training dataset), i.e., the learning dataset is divided into packages (mini-batches) with the number of patterns N. An additional element accelerating the achievement of the global minimum of the objective function of the problem under consideration (signal recognition) was the use of training data normalization (and the transmitted signals between CNN layers) using the batch normalization method [57].

A batch normalization algorithm normalizes the input training patterns and successive signal vectors propagated between successive ANN layers [57]. The normalization introduced by the discussed algorithm reduces or even eliminates possible oscillations of the ANN error minimization process and possibility to become stuck in the local minimum, resulting from the fact that small changes in the weight vector in the layers preceding subsequent layers may cause large changes in the weight vectors in subsequent layers (the deeper has an ANN structure), the so-called exploding gradient [67] or too-small changes to the weight vector value that will cause the weight vector gradient to fade away on subsequent layers, the so-called vanishing gradient [67].

In addition to the abovementioned stabilizing properties of the learning process, the batch normalization algorithm normalizes (providing the mean value and variance of the training pack: 0 and 1, respectively) in such a way that the training dataset that the range of values taken by ANN is approximately not variable, which is another aspect that reduces the probability of the occurrence of oscillations of objective function between successive epochs during the learning process.

The normalized input vectors for the successive ANN layers are calculated in the following way:(7)$$\mu_{B}=\frac{1}{m}\;\overset{m}{\underset{i=1}{\sum }}x_{i},$$
(8)$$\sigma_{B}^{2}=\frac{1}{m}\;\overset{m}{\underset{i=1}{\sum }}{\left(x_{i}-\mu_{B}\right)}^{2},$$
(9)$$\hat{x_{i}}=\frac{x_{i}-\mu_{B}}{\sqrt{\sigma_{B}^{2}+\epsilon }},$$
(10)$$y_{i}=\gamma \hat{x}+\beta =BN_{\gamma ,\;\beta }\left(x_{i}\right),$$
where B is the number of the current training packet (mini-batch [57] number), m is the number of elements in the packet, $\mu_{B}\;$ is the average value of the training packet, $\sigma_{B}^{2}$ is the variance of the teaching packet, $\hat{x_{i}}$ is the normalized input training vector, and $y_{i}\;$ is the normalized and scaled training vector. Algorithm 1 presents the exemplary of batch normalization procedure for an input vector X.
**Algorithm 1** Batch normalization procedureInput: X[N] ,
▷Input vector    N,▷Size of X    γ, β,▷Parameters to be learned    
ϵ
▷Very small value to stabilize calculation**Output**: Y[N]▷Normalized vector X1: **initialize**: $i=0,\mu_{B}=0,\;\;\sigma_{B}^{2}=0,\;s=0,x=0$

2: **for** i=0 **in** N−1 **do**▷Calculate mean of X3:  $\mu_{B}=\mu_{B}+X\left[i\right]$

4: **end for**
5: $\mu_{B}=\mu_{B}/N$

6: **for** i=0 **in** N−1 **do**▷Calculate variance of X7:  $s=X\left[i\right]-\mu_{B}$

8:  $\sigma_{B}^{2}=\sigma_{B}^{2}+s\cdot s$

9: **end for**
10: $\sigma_{B}^{2}=\sigma_{B}^{2}/N$

11: **for** i=0 **in** N−1 **do**▷Normalize X vector as Y vector12:  $x=\frac{X\left[i\right]-\mu_{B}}{\sqrt{\sigma_{B}^{2}+\epsilon }}$

13:  Y[i]=γx+β

14: **end for**


In the first step, the normalizing algorithm calculates the mean value $\mu_{B}$ and the variance $\sigma_{B}^{2}$ of the presented training set (Equations (7) and (8)); then, based on these values, it calculates the normalized and scaled input vectors $y_{i}$ of the training set (Equations (9) and (10)), [57]. The operation described above is used not only for the input layer, but for each successive output layer in relation to the next input layer (Figure 10) of the entire CNN structure.

## 5. Simulation Environment

In order to test the designed structures of neural networks, a simulator was created using C++ [68,69] language, with the FFTW [70] library for signal processing and the OpenCL [71] library to speed up the process of generating signals. The simulator enables one to generate many digital signals, introducing interferences and noises into them. The operation of the simulator is based on the generation of signals on a one-time base (vector of signal samples of length *N*), which consists of *N* samples, and all available signal classes with variable parameters PD, PRI, RF and SP. These signals parameters vary from generation to generation of successive signals in the given class. The simulator works in a quasi-real mode and allows one to view the currently generated signals, the modulus of their spectrum in the frequency domain, and the vector of samples of the signal space (Figure 11).

In the first step, the simulator reads the signal parameters for the given radar devices from the configuration file. Then it should check whether the given hardware configuration allows one to process signals with the maximum sampling frequency obtained from the configuration file. If the system‘s maximum sampling rate is less for random access memory (RAM) than the highest-frequency signal of all signal classes, then the signals (their waveform form) are scaled to the system’s acceptable sampling rate. For faster learning and processing purposes, the scaling rate was chosen manually. After calculating these scaling factors (or setting them manually), the simulator creates the filters with a finite impulse response (FIR) [72,73] for each radar signal class. Filter coefficients for individual classes were calculated based on the Hann‘s time window [74] using the formulas presented in Equations (11)–(13).
(11)$$c\left[i\right]=\{\begin{array}{cc} \mathrm{sinc}\left(x\left[i\right]\right) & dla\;x>0\\ 1 & dla\;x\leq 0 \end{array},\;\;sinc\left(x\right)=\frac{\sin\left(x\pi \right)}{x\pi },\;\;$$
(12)$$x\;\epsilon \left\{-l_{B}+s\cdot 0,-l_{B}+s\cdot 1,\ldots ,l_{B}+s\cdot \left(N-1\right)\right\},$$
(13)$$h\left[i\right]=c\left[i\right]\cdot \frac{\left(1-cos\left(\frac{2\pi i}{N}\right)\right)}{2},$$
where N is the number of FIR filter coefficients, x is the vector of discrete argument values for which the FIR filter coefficients are calculated, s is the discrete shift value between the sample x[i] and x[i+1], c is the vector of coefficients calculated according to the sinc(x) relationship, h is the vector of coefficients using the relationship for the Hann window, i is the successive index of the FIR filter coefficient and i ∈ (0, N−1), $l_{B}$ means the cut-off value of the FIR filter corresponding to the upper and lower cut-off frequency. The block diagram of simulation and recognition radar signals is presented in Figure 12.

After the FIR filters are created, the main simulation loop follows. The operation of the simulation loop consists of the following four steps described below.

The first step is the digital generation of signals on the one vector of the output signals S of real numbers with the number of elements $N=F_{max}$ (number of samples). A single-class signal is added to the S vector at intervals depending on the PRI. Initially, the signal vector contains approximately $K=\frac{1}{PRI\left[s\right]}$ of the waveforms of a signal for the given class, where the exact number depends on the drawing of individual values of PRI and PD in the successive signal generations for the given class. The values of PRI, PD and RF at each successive signal generation are randomly selected in accordance with the uniform distribution or PRI, with PD inter-pulse modulation in intervals characterizing the allowable range of signal parameter changes for a given class of signals. Below, Algorithms 1–3 in the form of a pseudocode are presented, describing the process of adding subsequent signals to the output vector of signals S.
**Algorithm 2** Add signals to vector space (Random PRI, PD, RF Modulation)**Input:**L =18▷number of signal classes   
N
▷number of samples for output vector of signals   
Cs[L]
▷L-Length vector of signals structures parameters   
ud(minValue, maxValue)
▷function to create uniform range distribution for random engine   
rg
▷random engine generator**Output**: S[N]▷N-Length output vector of signals1: **initialize**: i=0         sw 
▷vector of i-th time waveform signal2: **for** i=0 **in** (L − 1) **do**
3:  cl=Cs[i]
▷get parameters of i-th class of signal4:  priRange=ud(cl.pri_min, cl.pri_max)
▷create range distribution for PRI parameters5:  pdRange= ud(cl.pd_min, cl.pd_max)
▷create range distribution for PD parameters6:  rfRange = ud(cl.rf_min, cl.rf_max)
▷create range distribution for RF parameters7:  shift=0

8:  msl = cl.pd_max ⋅N
▷calculate max length of current class signal9:  **while**
((shift+msl)<N) **do**
10:   cPRI= priRange(rg)
▷get random value from PRI range of cl-signal11:   cPD= pdRange(rg)
▷get random value from PD range of cl-signal12:   cRF= rfRange(rg)
▷get random value from RF range of cl-signal13:   cWL=currentPD⋅N
▷get length of time waveform (TW) signal14:   sw=generate(cPD, cRF)
▷generate signal with PD and RF15:   AS2VS(S, sw, cWL)
▷Add signal to vector space, Algorithm 316:   shift=shift+cPRI⋅N
▷shift vector signal about shift value at output vector signal17:  **end while**
18: **end for**


**Algorithm 3** Add signal to vector space (random PRI modulation)
**Input:**

S[N]


▷N-Length output vector of signals
   
signalWaveform[currentWaveformLength]

▷number of samples for output vector of signals
   
currentWaveformLength

▷length of generated signal
   shift
▷shift at output signal vector space
**Output**: S[N]
▷N-Length output vector of signals
1: **initialize**: i=0

2: **for** i=0 **in** (currentWaveformLength − 1) **do**
3:  S[i+shift]=S[i+shift]+signalWaveform[i]

▷Adding signal to S vector space
11: **end for**


The second step is to introduce the interference and noise to the vector of the output signals. The noise introduced is additive. It is worth mentioning that the signals from different classes added to the same sample vector disturb each other by interfering with each other.

The third simulation step is the filtering step of the entire resulting vector from the output signals S, each of the 18 filters, and each of them is assigned to a given class signal. The block diagram of the output vector filter for each class of signals is shown in Algorithm 4.
**Algorithm 4** Filter all signals**Input:** S[N]▷N-Length output vector of signals   
L =18
▷number of signal classes   
filtersFIR[L]
▷L-Length vector of FIR filters1: **initialize**: i=0      filteredSignalVector[N]
▷N-Length output vector of i-th signal after filtration2: **for** i=0 **in** (L − 1) **do**
3:  firFilter =filtersFIR[i]
▷get i-th FIR filter4:  outputSignalVector = firFilter.filter(S)
▷try to filter i-th signal from S[N]5:  DNN_isClassSignal(outputSignalVector)
▷use DNN model to classify filtered signal11: **end for**


As can be deducted from the above-presented algorithms in the form of pseudocode, an output vector is given as an argument for each of the 18th FIR filters. The filtering function of the firFilter instance of FIR class returns the signal vector filtered against the i-th class, which is then sent to the CNN input, and an attempt is made to detect the given signal class. DNN_isClassSignal is a function which analyzes the occurrence of a given signal class and returns the probability value in the situation where the recognized signal is located in the given input vector.

## 6. Experiment Results

The calculation results of the effectiveness of radar signal recognition based on the post-detection PRI, PD determination and the sampled time form of the signal are presented in Table 7, Table 8, Table 9, Table 10, Table 11 and Table 12.

Where the disruption level means the amplitude level of the interfering signals.

The characteristics of changes in the probability of the correct recognition for 18 signal classes are presented in Figure 13, Figure 14 and Figure 15 on the basis of a separate analysis of the PRI, PD parameters and TW vectors and depending on the changes in the number of iterations of the training algorithm and the size of the training dataset.

The characteristic in Figure 13 indicates that teaching the CNN to recognize the correct class on the basis of PD parameter alone is unlikely: about 10% of the achieved effectiveness with a training dataset of 40–70 elements. This is due to large overlapping of the signal operation intervals for the PD parameter.

In the case of CNN analysis for the PRI parameter (Figure 14) in relation to the PD analysis (Figure 13), the results achieved are much better: approximately 70% of the effectiveness of the training package with the size of 20 elements and 60 epochs in the learning process. With larger packages at the level above 70, the quality of learning and CNN performance showed a descending trend.

The analysis of the results of the CNN recognition of TW samples (Figure 15) in terms of effectiveness was similar to the effectiveness of CNN in the PRI samples analysis. However, here the downward trend along with the growth of the learning dataset is much more noticeable. The optimal learning point turned out to be a package of 20 learning patterns and 140 epochs of the learning process. The combination of three CNN structures with the concatenation operation and an additional classifier with dense connections (Figure 7) which analyzes the combined tensor allowed one to achieve an efficiency of radar signal classes recognition at the level of 90–99% (Figure 16).

An important parameter that influences the qualitative operation of the structure (Figure 7, Table 4) is the size of the input vector accepted by the CNN. In this case, the size of CNN inputs, which analyzed the PRI and PD parameters, changed at the same time. The characteristics clearly indicate that the possibility of accepting a larger vector of samples (longer observation time) improved the efficiency of signal classification. The size of the package also turned out to be important. In this case, it should be at the level of about 350 elements for the CNN to achieve high effectiveness in the learning process.

The last examined structure (Table 5, Figure 8) contains efficiency of classification when the raw unprocessed sampled signal is considered. The characteristics of CNN operation in the case of disturbances are presented in Figure 17. The results of CNN operation without interference were at the level of 99% probability of correct recognition, therefore the tables and graphs with the operation of the CNN without interference were omitted.

The blue curve in Figure 17 shows the effectiveness of the CNN in the case of recognizing radar signals when there are a certain number of radio sources at different distances around the reconnaissance station. The noise levels (horizontal line) illustrate the signal amplitude level of interfering sources. The interfering signals, in this case, were signals for the 18 classes of radar signals examined earlier. To simulate the disturbance effect, the generated signals for the 18 classes to be recognized were superimposed on the remaining signals with a given amplitude (0.1–0.9). The red curve shows the disturbance noise.

A CNN working on the raw signals showed a high resistance to the noise of interference. However, it started to cope worse in the case of interference with other signals where the probability of correct recognition tended to decline along with the increase in the amplitude of the disturbing signals. To overcome this problem, the observation time (the number of samples received by the CNN at the input) was tested and increased, which resulted in an effectiveness equal to 92.2% (Table 11).

## 7. Conclusions

On the basis of the obtained results of the effectiveness measurements, it is possible to note the high effectiveness of the CNN in the process of recognizing digital signals when they are analyzed post-detection based on the PRI, and PD parameters and the form of the time waveform (RF and PD), or when we analyze the raw signal (not processed sampled signal). Simultaneous analysis of the set of three signal parameters was possible thanks to the concatenation operation of the three CNN network models (PD, PRI, TW), and the analysis of the response tensor obtained by the ANN with the dense full-connected architecture.

The achieved probabilities of correct signal recognition were high, ranging from 90–99%. However, in order to achieve such network efficiency in the case of post-detection analysis, it is required that the CNNs analyze more than two parameters of the radar signal. Otherwise, if each signal parameter is analyzed separately, then the radars’ signals cannot be properly classified. This is due to the overlapping of the ranges of the operating parameters of the classes of individual signals (Table 2), and was confirmed by the results of the signal recognition efficiency by the CNN (Table 7, Table 8 and Table 9 and Figure 13, Figure 14 and Figure 15), where the analysis of the parameters of each separately allowed one to achieve the maximum CNN values of the appropriate probability diagnosis at the level of 70–72% for PRI and TW, and a maximum of 11–16% for the PD parameter.

The CNN designed has the ability to classify signals on the basis of the analysis of raw data, also in the presence of interference at a 92–99% probability of correct recognition (Table 11 and Table 12). In the case of working with interference, the effectiveness of our CNN largely depends on the capabilities and sensitivity of the receiver, i.e., the ability to process signals such as a sampling frequency. Parameters such as package size and number of iterations (epochs) of the training algorithm were important for convergence by a particular CNN, and depend on both the architecture of the given CNN and the training dataset (Table 7, Table 8, Table 9 and Table 10) and (Figure 13, Figure 14, Figure 15 and Figure 16).

Increasing the size of the CNN input vector, which is basically the observation time or sampling rate, significantly improved the performance of the combined CNN (Table 4, Figure 7) and the CNN analyzing the raw signal (Table 5, Figure 8), especially in the event of disturbances (Table 11).

The results of the operation presented here were achieved in a relatively short time; about 2 h for a single learning process for post-detection analyzing networks, and about 24 h for a network working on the raw signals thanks to the scaling of high-frequency signals to lower frequencies. In the case of direct operation at high frequencies, changing the size of the CNN inputs and selecting an appropriate number of convolutional filters will be required. The learning time available and having the appropriate hardware architecture to carry out such a learning process should also be taken into account.

## Figures and Tables

**Figure 1 sensors-21-08237-f001:**
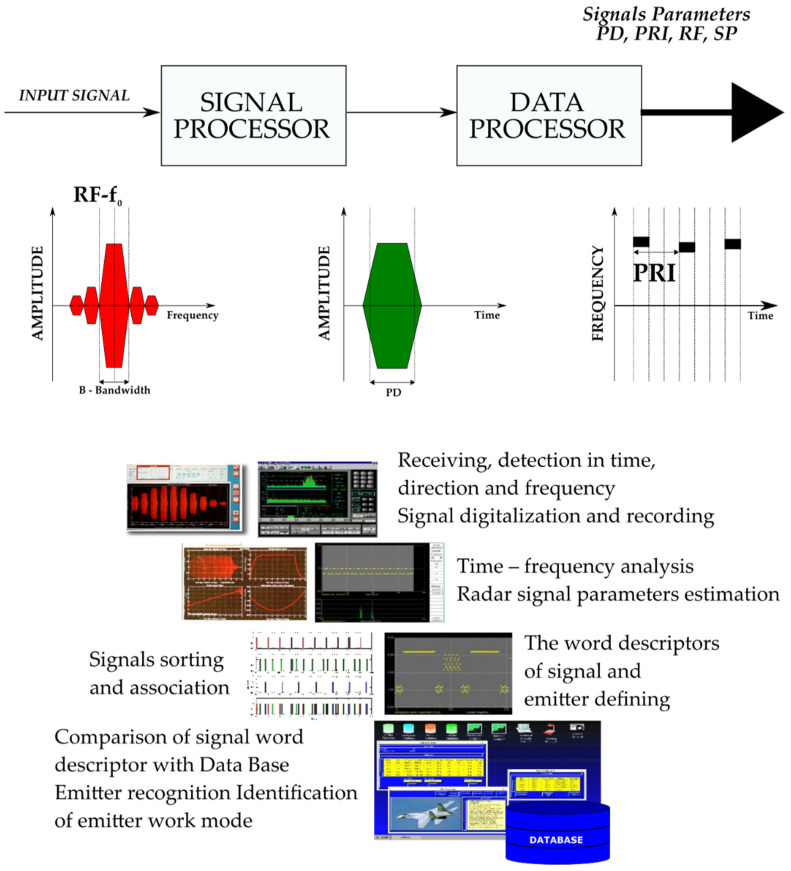
General structure of ESM system for measurement, recording, analysis and radar signals recognition [53].

**Figure 2 sensors-21-08237-f002:**
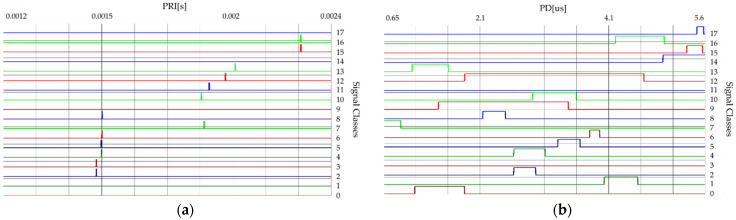
The confidence intervals of 18 classes of radars: (**a**) for parameter PRI; (**b**) for parameter PD.

**Figure 3 sensors-21-08237-f003:**
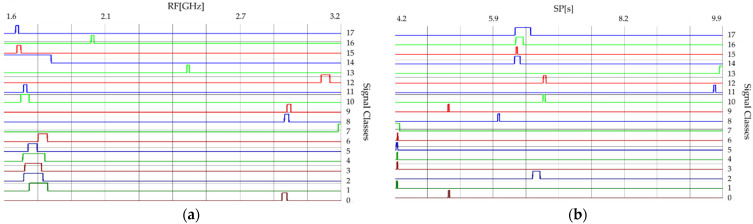
The confidence intervals of 18 classes of radars: (**a**) for parameter RF; (**b**) for parameter SP.

**Figure 4 sensors-21-08237-f004:**
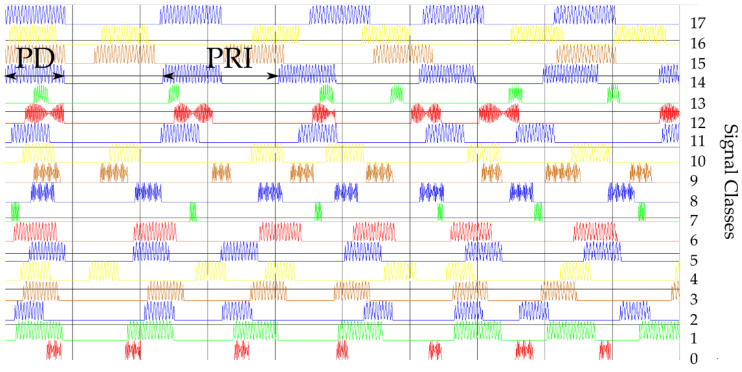
Examples of generated waveforms for the parameters PD and RF for 18 classes.

**Figure 5 sensors-21-08237-f005:**
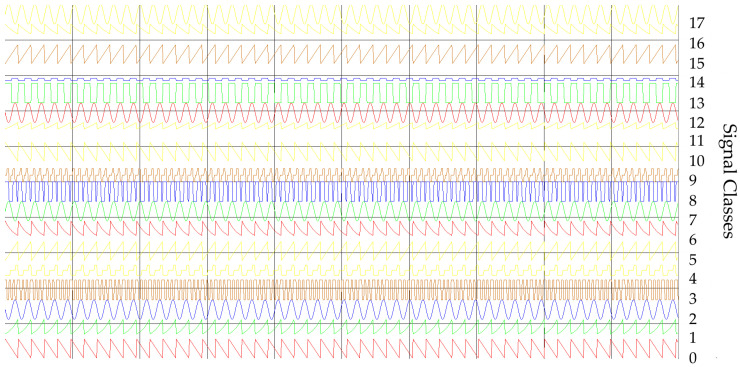
Illustration of training signals with inter-pulse modulation of PRI.

**Figure 6 sensors-21-08237-f006:**
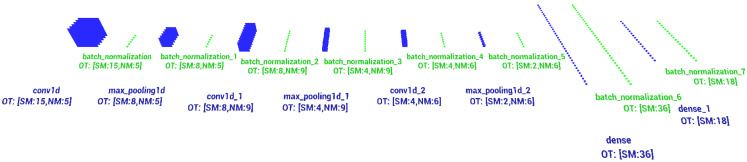
Structure of the CNN used for radar signal recognition, where OT is the output tensor, SM is the size map or number output of the dense layer, and NM is the number of maps.

**Figure 7 sensors-21-08237-f007:**
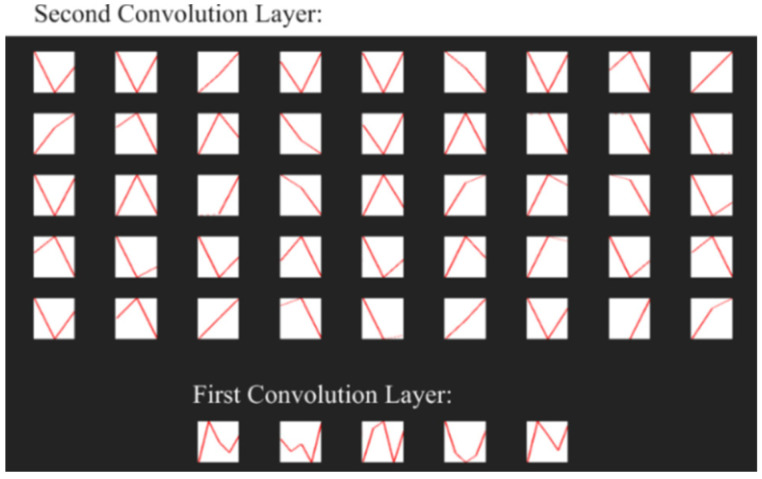
Example of obtained weight maps applied in the designed CNN.

**Figure 8 sensors-21-08237-f008:**
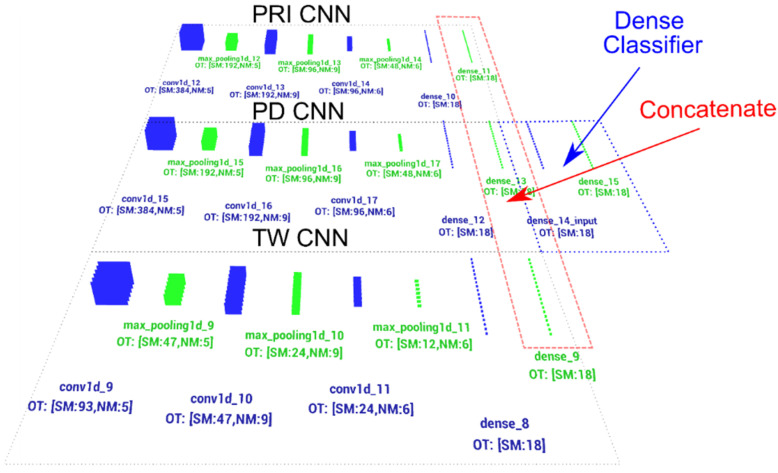
CNN structure for radar signals recognition based on the simultaneous analysis of PD, PRI and TW (RF + PD) parameters.

**Figure 9 sensors-21-08237-f009:**
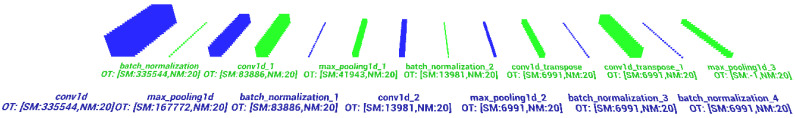
Structure of the CNN for radar signals recognition based on the analysis of the raw signal (raw samples of radar signal parameters).

**Figure 10 sensors-21-08237-f010:**
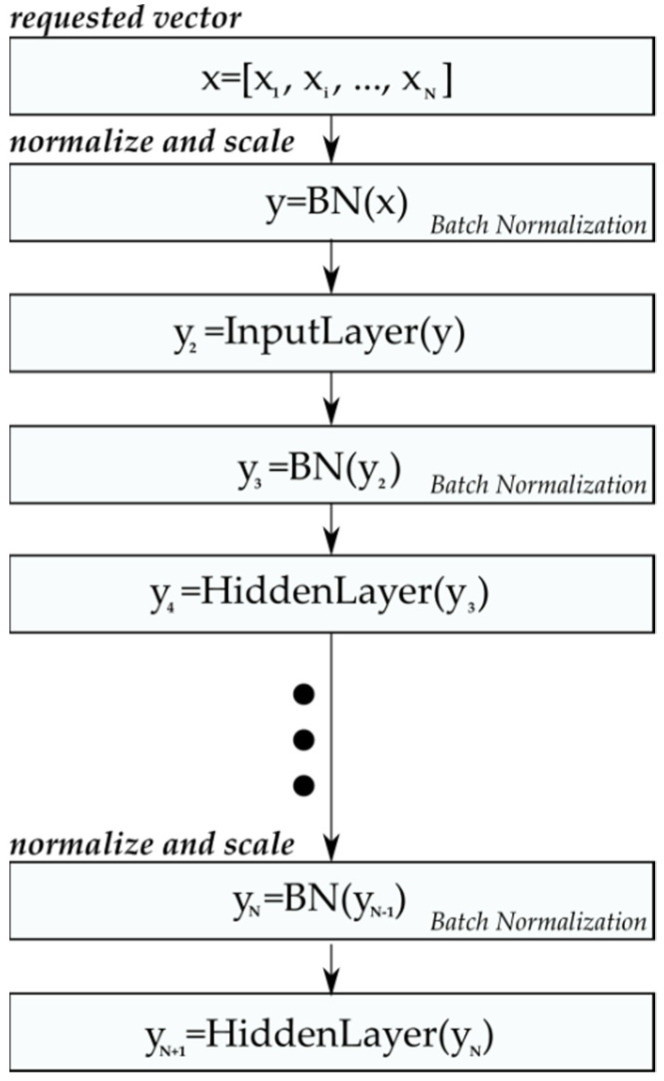
Scheme of normalizing the input vectors for successive layers using the batch normalization algorithm.

**Figure 11 sensors-21-08237-f011:**
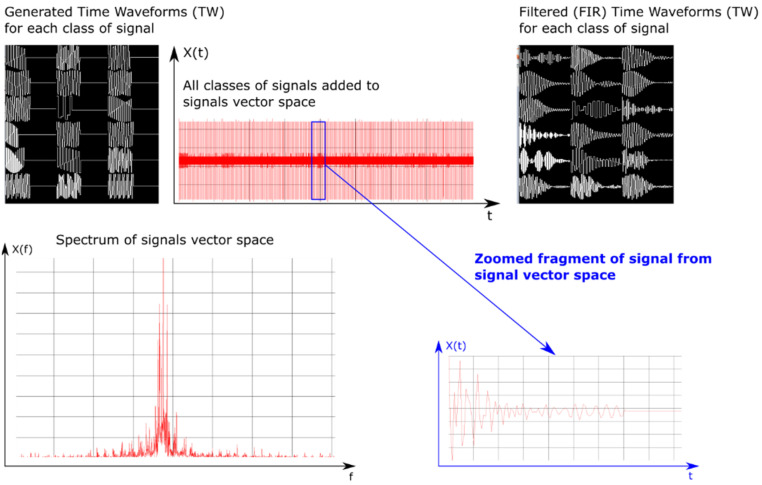
Example of the generation of signals parameters.

**Figure 12 sensors-21-08237-f012:**
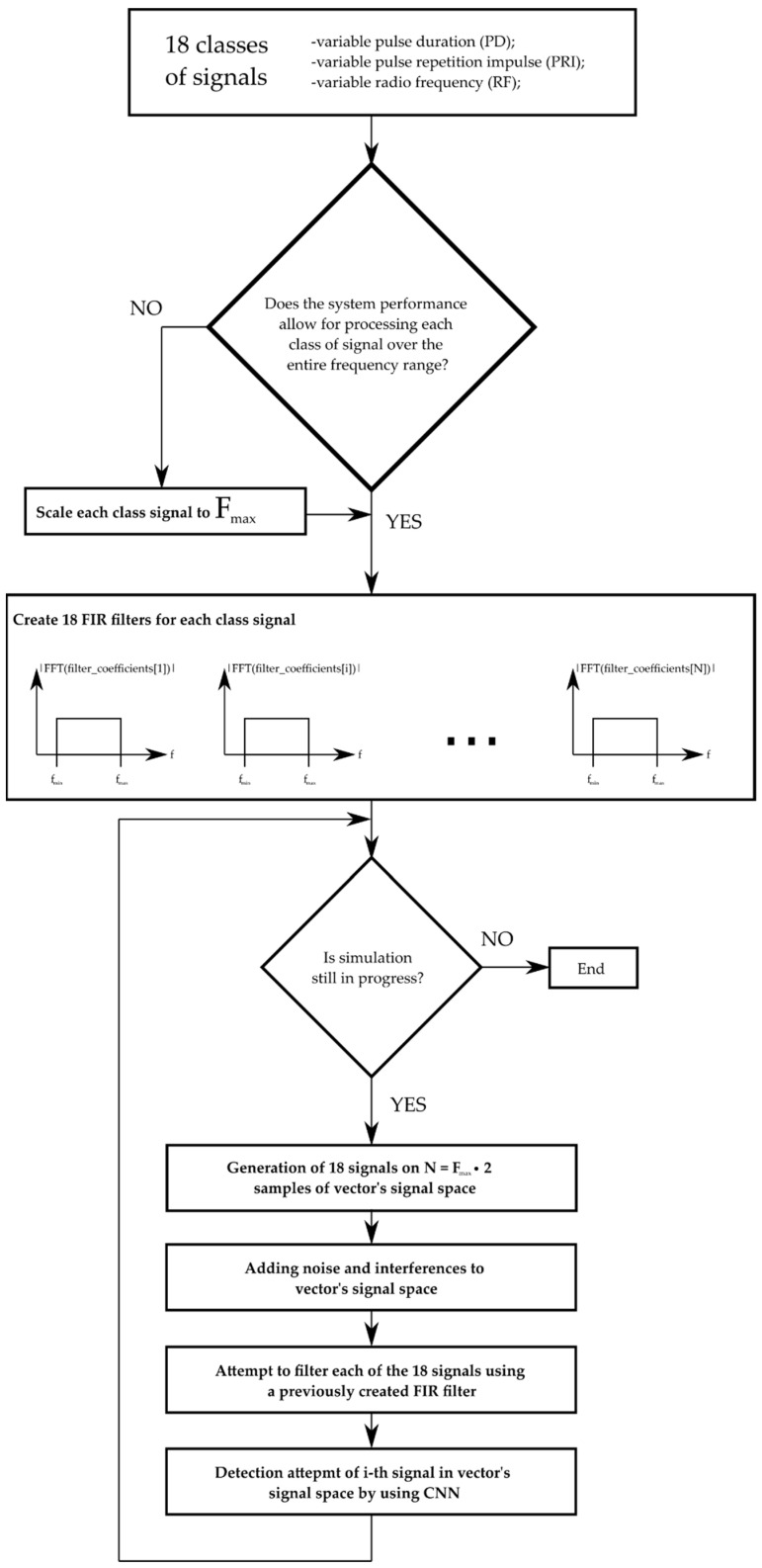
Diagram of the simulation and recognition of radar signals.

**Figure 13 sensors-21-08237-f013:**
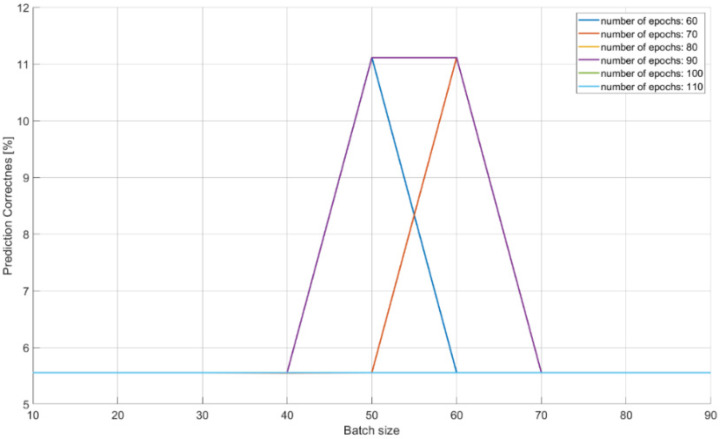
Effectiveness of radar signals recognition on the basis of the PD parameter in the function of the changes in the size of the training dataset and the number of iterations in the training algorithm.

**Figure 14 sensors-21-08237-f014:**
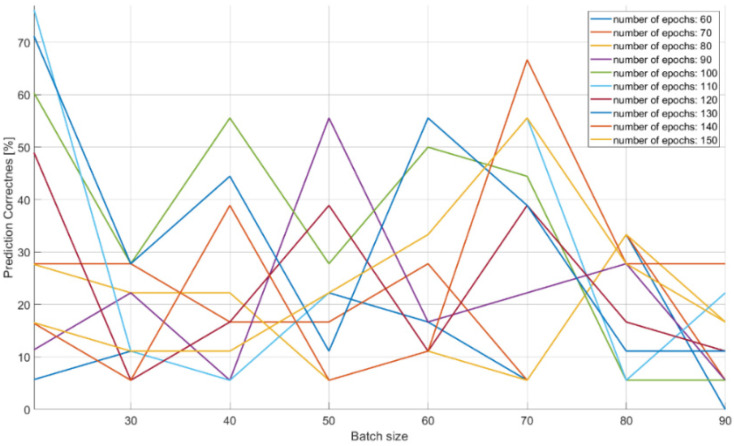
Effectiveness of radar signals recognition on the basis of the PRI parameter in the function of the changes in the size of the training dataset and the number of iterations in the training algorithm.

**Figure 15 sensors-21-08237-f015:**
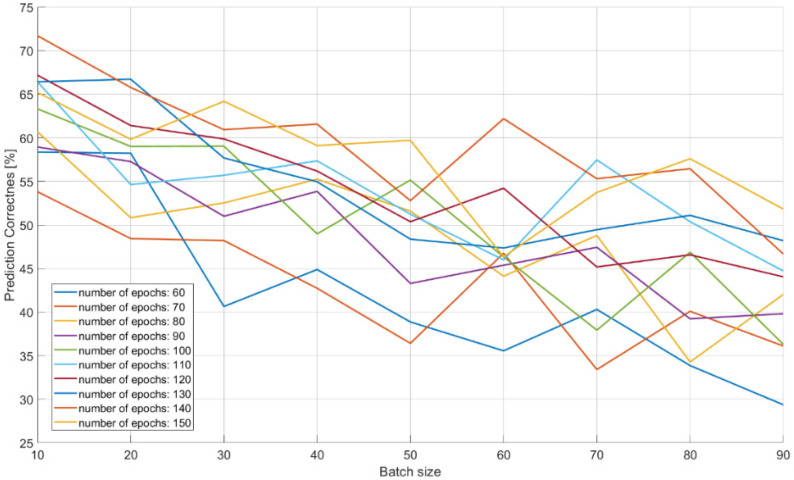
Effectiveness of recognizing radar signals on the basis of the time waveform shape modulation during the changes of the size of the training dataset and the number of iterations in the training algorithm.

**Figure 16 sensors-21-08237-f016:**
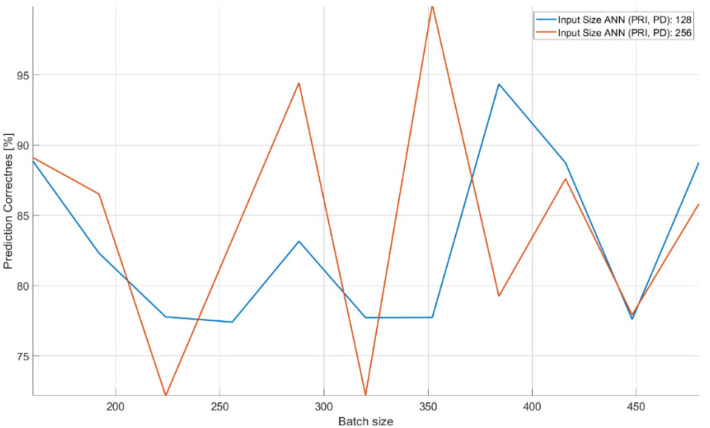
Effectiveness of recognizing radar signals on the basis of simultaneous analysis of three parameters (PD, PRI and TW) depending on the size of the training set and the number of iterations in the training algorithm.

**Figure 17 sensors-21-08237-f017:**
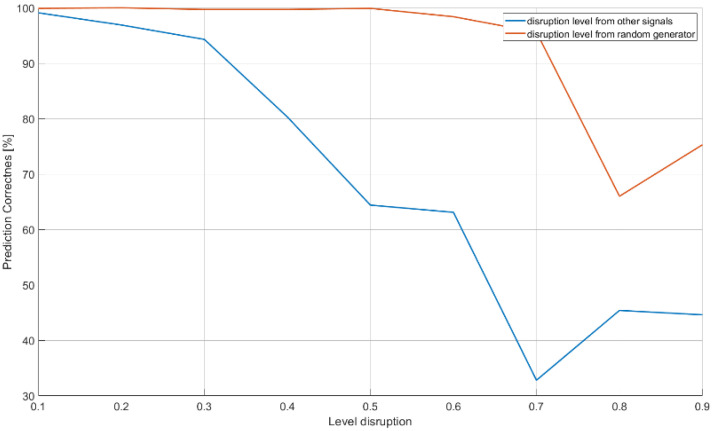
Effectiveness of signal recognition depending on the level of signal disruption.

**Table 1 sensors-21-08237-t001:** The confidence intervals of radar signal parameters for 18 classes examined.

Class Number	PRI [ms]	PD [μs]	RF [GHz]	SP [s]
0	0.877–0.878	0.929–1.725	2.800–2.832	3.97–4.00
1	1.229–1.230	3.958–4.492	1.255–1.368	2.85–2.87
2	1.223–1.223	2.512–2.863	1.221–1.339	5.80–5.96
3	1.223–1.223	3.277–3.277	1.228–1.330	2.86–2.88
4	1.248–1.250	2.512–3.015	1.215–1.351	2.86–2.87
5	1.247–1.250	3.216–3.571	1.248–1.303	2.85–2.88
6	1.250–1.252	3.727–3.885	1.308–1.365	2.87–2.88
7	1.751–1.752	0.431–0.705	3.144–3.162	2.81–2.92
8	1.251–1.252	2.018–2.379	2.816–2.842	5.04–5.08
9	0.768–0.768	1.308–3.384	2.832–2.854	3.96–3.99
10	1.738–1.739	2.811–3.514	1.203–1.254	6.02–6.07
11	1.775–1.778	3.482–3.482	1.220–1.240	9.72–9.76
12	1.856–1.858	1.727–4.592	3.040–3.092	6.03–6.09
13	1.905–1.905	0.888–1.466	2.219–2.235	9.85–9.91
14	2.150–2.150	4.898–5.570	1.100–1.389	5.41–5.53
15	2.225–2.228	5.280–5.529	1.180–1.205	5.44–5.47
16	2.224–2.226	4.138–4.917	1.633–1.650	5.43–5.59
17	2.375–2.375	5.440–5.548	1.171–1.190	5.42–5.76

**Table 2 sensors-21-08237-t002:** Overlapping of individual signals in PRI, PD, RF and SP parameters.

Number of Signal Class	Number of Overlapping Signals in PRI	Number of Overlapping Signals in PD	Number of Overlapping Signals in the RF	Number of Overlapping Signals in the SP
0	-	9, 13	8	9
1	-	12, 16	2, 3, 4, 5, 6, 14	3, 4, 5, 6, 7
2	3	4, 9, 10, 12	1, 3, 4, 5, 6, 10, 11, 14	-
3	2	5, 9, 10, 12	1, 2, 4, 5, 6, 10, 11, 14	1, 4, 5, 6, 7
4	5	2, 9, 10, 12	1, 2, 3, 5, 6, 10, 11, 14	1, 3, 5, 6, 7
5	4	3, 9, 10, 11, 12	1, 2, 3, 4, 10, 14	1, 3, 4, 6, 7
6	8	12	1, 2, 3, 4, 14	1, 3, 4, 5, 7
7	-	-	-	1, 3, 4, 5, 6
8	6	9, 12	0, 9	-
9	-	0, 2, 3, 4, 5, 8, 10, 12, 13	8	0
10	-	2, 3, 4, 5, 9, 11, 12	2, 3, 4, 5, 11, 14, 15	12
11	-	5, 10, 12	2, 3, 4, 10, 14	-
12	-	1, 2, 3, 4, 5, 6, 8, 9, 10, 11, 16	-	10
13	-	0, 9	-	-
14	-	15, 16, 17	1, 2, 3, 4, 5, 6, 10, 11, 15, 17	15, 16, 17
15	16	14, 17	10, 14, 17	14, 16, 17
16	15	1, 12, 14	-	14, 15, 17
17	-	14, 15	14, 15	14, 15, 16

**Table 3 sensors-21-08237-t003:** Detailed structure of designed CNN for separate signals recognition depending on the examined parameters (PRI, PD, TW).

Layer Number	Layer Type	Layer Dimension	Activation Function
0	B	-	-
1	R ^1^	-	-
2	C_1D ^2^	[KS ^3^: 5, NK ^4^: 5, SS ^5^: 1]	ReLU [20,62]
3	B ^6^	-	-
4	MP_1D ^7^	[SS: 2]	-
5	C_1D	[KS: 3, NK: 9, SS: 1]	ReLU
6	B	-	-
7	MP_1D	[SS: 2]	-
8	C_1D	[KS: 2, NK: 6, SS: 1]	ReLU
9	B	-	-
10	MP_1D	[SS: 2]	-
11	F	-	-
12	D ^8^	-	ReLU
13	B	-	-
14	Dropout	-	-
15	D	-	Softmax [20]
16	B	-	-

^1^ Reshape layer [3], ^2^ Convolution 1D layer, ^3^ Kernel size, ^4^ Number of kernels, ^5^ Size stride, ^6^ Batch Normalization layer, ^7^ Max Pooling 1D layer, ^8^ Dense layer.

**Table 4 sensors-21-08237-t004:** Detailed structure of CNN designed for radar signal recognition with simultaneous input of three signal parameters (PRI, PD, TW).

Input Vectors:			
PD Samples (Post-Processing)		PRI Samples (Post-Processing)	TW Samples (Raw Acquired Signal Samples)
Structure CNN for PD parameter ^9^		Structure CNN for PRI parameter	Structure CNN for TW parameter
**Associated Outputs in CNN for PD, PRI and TW Parameters**			
Layer number	Layer type	Dimensions of layer	Activation function
0	D	-	ReLU
1	B	-	-
2	D	-	Softmax
3	B	-	-

^9^ For parameters PD, PRI, TW, the structure described in Table 3 was used.

**Table 5 sensors-21-08237-t005:** Detailed structure of designed CNN for recognizing sampled signals.

Layer Number	Layer Type	Dimensions of Layer	Activation Function
0	R	-	-
1	C_1D	[KS: 5, NK: 30, SS: 1]	ReLU
2	B	-	-
3	MP_1D	[SS: 2]	-
4	C_1D	[KS: 4, NK: 30, SS: 2]	ReLU
5	B	-	-
6	MP_1D	[SS: 2]	-
7	C_1D	[KS: 3, NK: 30, SS: 3]	ReLU
8	B	-	-
9	MP_1D	[SS: 2]	-
10	C_1D	[KS: 4, NK: 30, SS: 2]	ReLU
11	B	-	-
12	C_1D	[KS: 5, NK: 30, SS: 1]	ReLU
13	B	-	-
14	MP_1D	[SS: 2]	-
15	F	-	-
16	D	-	Softmax

**Table 6 sensors-21-08237-t006:** The designed network parameters.

Processing Network	Input Tensor Size	Number of Layers	Number of Weights	Size on Disk	Processing Time for a Single Tensor [s]	GPU Processor
TW or PD or PRI	(1, 128)	17	2248	63 kB	0.015	Geforce 1060 GTX 6GB
TW + PRI + PD	(3, 128)	3 × 17 (In parallel) + 3 (Output) = 54	2248 × 3 + 1332 = 8076	(63 × 3 + 18) kB = 207 kB	0.046	

**Table 7 sensors-21-08237-t007:** Effectiveness of signal classification using CNN based on the PD vectors.

Input Size of ANN:			512		
Number of Samples ^10^:	190	Number of Tests ^11^:	1900	η^12^:	0.0001
Epochs ^13^ [B:S:E] ^14^		Batch Size [B:S:E]		Prediction ^15^ [Min–Max]	
60		10:10:90		0.056–0.111	
70		10:10:90		0.055–0.111	
80		10:10:90		0.056–0.056	
90		10:10:90		0.056–0.111	
100		10:10:90		0.056–0.056	
110		10:10:90		0.056–0.056	
110:10:240		90		0.056–0.167	

^10^ number of learning patterns, ^11^ number of testing patterns, ^12^ learning factor, ^13^ number of executed learning iterations, ^14^ [B:S:E] is the changes of parameters (B is the begin value, S is the step value, E is the end value), ^15^ effectiveness of the correct recognition of the presented signals from one of the 18 classes.

**Table 8 sensors-21-08237-t008:** Effectiveness of signal classification using CNN based on the PRI vectors.

Input Size of ANN:			512		
Number of Samples:	190	Number of Tests:	1900	η:	0.0001
Number of Epochs	Batch Size [B:S:E]			Prediction [Min–Max]	
60	10:10:90			0.000–0.333	
70	10:10:90			0.056–0.500	
80	10:10:90			0.056–0.333	
90	10:10:90			0.056–0.556	
100	20:10:90			0.056–0.611	
110	20:10:90			0.056–0.778	
120	20:10:90			0.056–0.500	
130	20:10:90			0.111–0.722	
140	20:10:90			0.056–0.667	
150	20:10:90			0.111–0.556	

**Table 9 sensors-21-08237-t009:** Effectiveness of signal classification using CNN based on the TW vectors.

Input Size of ANN:			93		
Number of Samples:	190	Number of Tests:	1900	η:	0.0001
Number of Epochs [B:S:E]		Batch Size [B:S:E]		Prediction [Min–Max]	
60		10:10:90		0.294–0.584	
70		10:10:90		0.334–0.538	
80		10:10:90		0.343–0.607	
90		10:10:90		0.392–0.589	
100		10:10:90		0.363–0.633	
110		10:10:90		0.447–0.664	
120		10:10:90		0.440–0.672	
130		10:10:90		0.474–0.667	
140		10:10:90		0.467–0.717	
150		10:10:90		0.463–0.652	

**Table 10 sensors-21-08237-t010:** Effectiveness of signal recognition using the CNN on the basis of simultaneous analysis of 2 parameters PRI, PD and TW vectors.

Input Size of ANN (TW):			93		
Number of Samples:	190	Number of Tests:	1900	η:	0.0001
Number of Epochs [B:S:E]		Input Size (PRI, PD) [B:S:E]		Prediction [Min–Max]	
128		128:32:288		0.774–0.889	
128		320:32:480		0.776–0.944	
128		512:32:672		0.808–0.944	
128		704:32:864		0.670–0.889	
128:32:160		896:32:928		0.778–0.889	
160:32:224		960:32:992		0.832–0.923	
256		128:32:288		0.722–0.944	
256		320:32:480		0.722–1.000	

**Table 11 sensors-21-08237-t011:** Efficiency of signal recognition using the CNN based on a sampled signal in the presence of interference from other signals.

Input Size of ANN:				$3.3554*10^{5}$			
Number of Samples	80	Number of Tests	80	Batch Size:	40	η:	$5*10^{-5}$
Number of Epochs		Prediction		Disruption Level			
256		0.991		0.1			
		0.969		0.2			
		0.943		0.3			
		0.803		0.4			
		0.644		0.5			
		0.631		0.6			
		0.328		0.7			
		0.454		0.8			
		0.446		0.9			
**Input Size of ANN:**				$8.3886\cdot 10^{5}$			
499		0.922		0.9			

**Table 12 sensors-21-08237-t012:** Efficiency of signal recognition using the CNN based on a sampled signal in the presence of pseudorandom noise.

Input Size of ANN:						$3.3554*10^{5}$			
Number of Epochs:	256	Number of Samples:	80	Number of Tests	80	Batch Size	40	η:	$5*10^{-5}$
Prediction						Disruption Level			
0.999						0.1			
1.000						0.2			
0.997						0.3			
0.997						0.4			
0.999						0.5			
0.984						0.6			
0.956						0.7			
0.660						0.8			
0.753						0.9			

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
