# Peer review of "Specific Radar Recognition Based on Characteristics of Emitted Radio Waveforms Using Convolutional Neural Networks"

_sensors, 2021, doi:10.3390/s21248237_

Round 1

Reviewer 1 Report

In this manuscript, the authors introduce a method to recognize radar signals using CNN.  But it seems that it needs major revision to be published.

It is not clear what the main idea of this paper is. As mentioned in the manuscript, radar signals can be categorized into 18 classes depending on PRI, PD, RF, and SP. The authors applied CNN method to classify radar signals. But, what is the advantage of using CNN? Why do we need CNN? Is it better than previous  methods? I think you have to justify using CNN by comparing the classification results of CNN and old methods. It should be the main topic. I don’t think it is necessary to compare the results of the CNN based on only one parameter with those based on combinations of two or three parameters. We can easily predict more parameters bring better results. We can also expect that having random noise is less severe to the performance than having interference signals. We would have the same conclusion even if we use other classification methods than the CNN method. Furthermore, it is not clear how the number of epochs and batch size affects the performance. I can not see any tendency. So, we can not get any conclusion from it. Authors did extensive simulations but it does not support the main idea of the paper. It means that large part of the chapt. 6 could be removed. 

The paper explains the simulation method too much. Generating radar signals and FIR filter coefficients in the manuscript are not so new approaches. It is less related to the main topic of the paper which is about correct recognition of the radar classes. You can omit or reduce the major part of the chapt. 5. Then the length of the paper would be much shorter than the current version.

The current manuscript just shows the classification results of the radar signals using CNN method with some variations in the simulation. In order to be published, it needs to show the advantages of the proposed CNN method over non-CNN methods and it should not contain basic simulation methods and results which does not contain any new findings or results.

Reviewer 2 Report

The authors present the application of CNN to classify up to 18 types of radar signals. Their classification results have good accuracy. However, it is not clear to me that why we need this type of classification for radar signals, where to apply their proposed framework? 

Reviewer 3 Report

Line 36, synthetic aperture radar can also offer a better decision-making, must add some related work of the balance scene learning mechanism.

Line 42, the reference citations should be in order.

Line 44, ANN has applied to objects recognition, but the authors ignore the hog-shipclsnet and pfgfe-net, please add it.

Line 48, Too references, e.g., [11, 13, 16, 17−19, 33, 35, 36], please try to avoid such.

Line 50, with respective to the broadly understood classification problem, the se-lpn-dpff can be considered.

Line 67, Chaper - > Section. This is an article, instead of a book.

Line 97, add the system parameters of Figure 1, e.g., the center f, and so on.

Line 138, Figure 6, what is the difference between the conv1d and the traditional conv in the signal processing field.

Line 225, you should add a table process of the BN, like the type in high-speed ship detection in sar images based on a grid convolutional neural network.

It will be welcome to add this work here. How about the network parameters? Is it lightweight? You can add some work in the future about this, e.g., shipdenet-20 and hyperli-net. The structure of the whole article is in a mess somehow. Please change it carefully. Good work. Accept after a major revision.

Round 2

Reviewer 1 Report

Thank you for your quick response and resubmitting of the revised manuscript. But, I am afraid I have to say that not all of my previous comments are addressed enough. 

  1. My major comment for the original manuscript was about why the authors used CNN to recognize radar signals. That’s why I asked what is the main idea of the paper and what is the advantages of using CNN over previous conventional methods. In the response no.2, it is said that ‘classification algorithms based on the neural networks in pattern recognition show much greater resistance than handwritten algorithms to all kinds of disturbances ….’ This is a very strong claim. I think it needs a reference and it needs to be shown in the simulation results. I asked to include performance comparison between the applied CNN method and conventional ones. Or, at least, you can show some specific examples where CNN outperforms the old ones. I am not saying that CNN is not good. As you mentioned, you may compare two methods with inter-pulse modulation signals to show the advantages of your method. I don’t think the advantages of CNN over fully connected NN is the main topic of the manuscript. But, the authors talk about general advantages of CNN over ANN which are not specifically related with radar signal recognition
  2. I also commented on the length of the paper recommending reduction of the signal generation part in the simulation session and simulation results which can be easily predicted. But, the authors refused. I think short paper which can present main idea in a compact form is a good one.
  3. Since my main comments are not addressed at all, I don’t think I can change my previous review result.

Reviewer 3 Report

Well done. Accept.

Round 3

Reviewer 1 Report

Thank you for your revised manuscript.  Good luck on your next research.